# Near-Field Single-Scattering Calculations of Aerosols: Sensitivity Studies

Nkongho Ayuketang Arreyndip *,† , Konrad Kandler † and Aryasree Sudharaj

Atmospheric Aerosol Group, Institute of Applied Geosciences, Technical University of Darmstadt, 64289 Darmstadt, Germany; kzk@gmx.de (K.K.); aryasree@geo.tu-darmstadt.de (A.S.)
* Correspondence: ayuketang@aims-cameroon.org
† These authors contributed equally to this work.

**Abstract:** We model the effects of the photosensitive parameters of aerosols on their optical properties to provide a solid framework for further experimental and theoretical studies. A spherical dust particle is used to study the effects of the ambient medium, size, surface roughness, wavelength, and imaginary part of the complex refractive index. Five Gaussian random spheres with different aspect ratios are simulated to study the dependence of aerosol light scattering properties on particle shape distribution. To investigate the influence of composition, we model two typical kaolinite-like particles (pure and composite) collected from Southwest Sahara, with 0 and 2% hematite at different mixing states. Using the method of discrete-dipole approximation in DDSCAT, a comparative study is performed with the Mueller matrix elements, scattering, absorption, extinction efficiencies, single scattering albedo, and linear depolarization ratio as indicators. For single, microscopic dust particles, near-field calculations are carried out. The results show that the intensity of backscattering and the scattering efficiency decreases in water compared to dry air. Light in the visible range is more efficient for aerosol scattering experiments. A small number of impurities in the sample will increase its absorbing properties, but, in general, the scattering efficiencies strongly depend on the single-particle mixing state. Smaller particles with a diameter comparable to the wavelength of incident light show higher scattering efficiencies but lower backscattering intensities than larger particles, while surface roughness is shown to strongly alter the polarizability of the particle but has a negligible effect on its single-scattering albedo. Moreover, different shapes have a strong effect on the degree of linear polarization, but, in general, using the spherical over elliptic shape model can underestimate the scattering efficiencies by up to 4%. Finally, variation in the imaginary part of the complex RI can underestimate the single scattering albedo by up to 35.8%.

**Keywords:** single-scattering; aerosol photosensitivity; near-field; modeling

## 1. Introduction

The increase in the concentration of aerosols in the atmosphere, especially mineral dust, under climate forcing can seriously upset the earth's radiation budget [1–4] via the emission of the absorbed radiation in the form of thermal radiation leading to more earth warming [3,5,6]. Increasing earth warming has equally been found to increase surface evaporation, thereby leading to an intensification of atmospheric water vapor (atmospheric rivers) [7]. This increase in humidity under anthropogenic forcing can cause the medium surrounding atmospheric aerosols to shift from dry air to water. This process may significantly affect the scattering properties of aerosol particles as water droplets can also significantly scatter light. Other photosensitive parameters such as size, shape, composition, surface roughness, and the imaginary part of the complex refractive index of the particle have been independently studied and found to affect aerosol optical properties. Aerosol light-scattering properties have equally been found to have many other applications such as investigating climate variability [8]; Lidar remote sensing; and environmental

monitoring [6]. Additonally, this scattering phenomenon can also significantly affect space optical communications through the distortion of laser beams, which can lead to a decrease in the quality of the transmitted signals [9,10]. Environmental engineers have also found this scattering phenomenon to be useful in air quality monitoring and industrial hygiene [11,12].

The impacts of these photosensitive parameters on aerosol scattering properties have been at the center of both the theoretical and experimental research [4,6,13–17]. The shape of aerosol particles plays a major role in their interaction with electromagnetic radiation, affecting mainly lidar, satellite, and spectroscopic measurements of forward and backward scattering [18,19], as well as measurements of aerosol size distributions. Particle shape has been assumed to influence radiance but not the radiative energy balance and, subsequently, climate [20]. However, recent investigations have shown that particle non-sphericity has a significant impact on radiative energy transfer [13,19,21]. Additionally, Yang et al. [22] have shown that a shape-dependent separation of mineral dust into layers of different altitudes during cross-Atlantic transport of Saharan dust exists, which strongly points to a shape-preferential settling of particles. By using direct measurements of shape and chemical compositions instead of mathematical shape models, Lindqvist et al. [6] simulated the light scattering by single, inhomogeneous mineral dust particles. Their main finding was that particle-to-particle variation in scattering by mineral dust was quite considerable and was not well reproduced by simplified shapes of homogeneous spheres, spheroids, or Gaussian random spheres.

Besides the general particle shape, surface roughness is currently practically unaccounted for, though it has a considerable impact on the particle light scattering properties [20]. The mixing state of dust particles has also been found to have an impact on the atmospheric optics [20,23] as well as on the cloud-forming properties of the dust [24]. Zieger et al. [14] experimentally investigated the impact of atmospheric humidity on the aerosol shape and scattering properties. They found that aerosol particles experience hygroscopic growth in the ambient atmosphere and that their optical properties are therefore strongly dependent on the ambient relative humidity (RH). Cynthia et al. [25] equally investigated the effect of changes in relative humidity on aerosol scattering near clouds using in situ aircraft observations. They found that a relative humidity increase is sufficient to cause substantial growth of hygroscopic aerosol particles and consequently greatly enhance particle scattering cross-sections near clouds. Meanwhile, Huang et al. [5] investigated the importance of the dust refractive index for the development of a model of dust optical properties. Their main finding was that the scattering matrix elements of different kinds of dust particles are reasonably reproduced by choosing appropriate RIs, even when using a fixed particle geometry. By using the Mie theory to estimate the complex refractive index, Biagio et al. [26] presented a new dataset of complex refractive indices and single-scattering albedos (SSA) for 19 mineral dust aerosols over the 370–950 nm range in dry conditions. They found that the imaginary part of the refractive index (k) and the SSA vary widely from sample to sample assuming that all particles are spherically shaped. This, together with the findings by Zong et al. [4], points to the fact that the wavelength of the incident light is a major factor when performing optical calculations to determine the imaginary part of the complex refractive index of the dust.

Determining the optical properties of dust aerosols is a big challenge, especially through experiments. However, researchers have been able to carry out theoretical calculations with parameters obtained directly from in situ measurements [6]. The most established theoretical approach is making use of the discrete dipole approximation (DDA). The DDA is a classical numerical technique used for computing light scattering by particles of arbitrary shapes. The DDA was first introduced by Purcell et al. [27] for modeling light-scattering by arbitrary shapes, compositions, and sizes (with sizes in the range from nanometers to microns [28]). Draine and Flatau [29–32] have further developed and updated the discrete dipole scattering (DDSCAT), an open-source software package that runs on DDA, written in Fortran. Draine and Flatau have used this software to compute scatter-

ing and absorption by isolated, homogeneous spheres as well as by targets consisting of two contiguous spheres. They have equally extended the theory to include periodic targets with the possibility of performing fast near-field calculations for regular rectilinear grids [32]. Other open-source packages have been developed with the DDA and have been found to be robust for carrying out optical calculations such as Amsterdam DDA (ADDA) [33–35], as well as DDA-SI, a computation toolbox written in Matlab [28]. The MOPSMAP package (modeled optical properties of ensembles of aerosol particles), which is computationally fast for optical modeling even in the case of complex aerosols, has recently been developed [36].

Our goal in this work is to simulate, using data from in situ measurements, the impacts of the major photosensitive parameters of aerosols on their optical properties to generate results that can act as a reference in the field of aerosol optics. We employ the most recent version of DDSCAT (DDSCAT 7.3), developed by Draine and Flatau [37] for sensitivity studies. This version of DDSCAT can be used to model light-scattering by regular shapes such as spheres, spheroids, rectangular, cylinders, ellipsoids, and prisms, as well as arbitrary shapes, clusters of particles, and periodic targets. Additionally, scattering by isotropic and anisotropic materials can be modeled, while near electric and magnetic field calculations can equally be performed for different orientations of the target. In DDSCAT, the particle's volume is replaced by a set of point dipoles in 3D. Pure and composite materials can be modeled by assigning the same or different complex refractive indices to these dipoles. The input to DDSCAT requires one to specify the wavelength of the incident light, the shape file of the particle (3D dipoles), the complex refractive index of the particle, the refractive index of the ambient medium, the fast Fourier transform (FFT) method, the Mueller matrix elements to be printed, the target orientation to incident light, etc, while the output generates information about the computed Mueller matrix elements, scattering, absorption, and extinction coefficients, averaged over the different orientations and polarization states. The single scattering albedo and linear depolarization ratios are computed from mathematical relations given in Equations (9) and (11) of Section 2.1, respectively. To investigate the effects of each photosensitive parameter on dust optical properties, the imaginary part of the complex refractive index is varied, the refractive index of the ambient medium is varied, different light sources are used, and different particle shapes and sizes are tested. Pure and composite materials together with their different single-particle mixing states are compared. The Mueller matrix elements, scattering, absorption, extinction efficiencies, single scattering albedo, and linear depolarization ratios are used as indicators.

The remaining sections are as follows: In Section 2, we will briefly revisit the theory behind the DDA and describe in detail the modeling we performed in DDSCAT-7.3. Section 3 is dedicated to the results of our findings, followed by a discussion in Section 4. We conclude our findings in Section 5.

## 2. Method

Most scattering problems will require the computation of the scattering matrix **S**, which describes the angular distribution of the intensity and polarization of the scattered radiation. The incident and scattered radiation of the Stoke vector $[I, Q, U, V]^T$ are linked through the scattering matrix **S** [6,37,38], where $I$ denotes the intensity and $Q$, $U$, and $V$ the polarization states. The relationship between the incident and scattered radiation is therefore given by [6]

$$\begin{pmatrix} I_s \\ Q_s \\ U_s \\ V_s \end{pmatrix} = \frac{1}{k^2 r^2} \begin{pmatrix} S_{11} & S_{12} & S_{13} & S_{14} \\ S_{12} & S_{22} & S_{23} & a_{24} \\ -S_{13} & -S_{23} & S_{33} & S_{34} \\ S_{14} & S_{24} & -S_{34} & S_{44} \end{pmatrix} \begin{pmatrix} I_i \\ Q_i \\ U_i \\ V_i \end{pmatrix} \qquad (1)$$

The $S_{ij}$ matrix is also called the Mueller matrix. The Mueller matrix depends on the wavelength of incident light $\lambda$; the geometry of the target; and other physical properties of the target such as the shape, size, and composition. The $S_{11}$, $S_{21}$, $S_{31}$, and $S_{41}$ Mueller matrix

elements describe the intensity and polarization state for the scattering of unpolarized incident radiation. The polarization $P$ of the scattered light is given by [37]

$$P = \frac{(S_{21}^2 + S_{31}^2)^{1/2}}{S_{11}}. \tag{2}$$

The value $S(\theta) = S_{11}(\theta)$ gives the scattering phase function, while $Q(\theta) = S_{12}(\theta)$ gives the Polarized phase function where $\theta$ is the scattering angle. The anisotropy of the scattered light is described by the asymmetric parameter $g$ given by

$$g = \langle \cos \theta \rangle = \int_{4\pi} \cos \theta \frac{S_{11}(\theta)}{4\pi} d\Omega \tag{3}$$

where $g$ lies between +1 and −1. When $g$ is either 1, −1, or 0, we have forward, backward, or isotropic scattering, respectively. In most applications, six independent Mueller matrix elements ($S_{11}$, $S_{12}$, $S_{22}$, $S_{33}$, $S_{34}$, and $S_{44}$) are usually calculated when we consider a large number of particles and their mirror particles.

### 2.1. Calculated Parameters

The differential scattering cross section can be obtained directly from the Mueller matrix by the expression [37]

$$I_s = \frac{1}{r^2} \left( \frac{dC_{scat}}{d\Omega} \right)_{s,i} I_i \tag{4}$$

where $C_{scat}$ is the scattering cross-section. In this work, we consider a linearly polarized light with Stokes vector $s_i = I(1, 1, 0, 0)$ where the incident electric field $E_i$ is parallel to the scattering plane. The total scattering cross-section is given by [37]

$$\frac{dC_{scat}}{d\Omega} = \frac{1}{k^2} (S_{11} + S_{12}) \tag{5}$$

So, for the case where $S_{11} >> S_{12}$,

$$\frac{dC_{scat}}{d\Omega} \approx \frac{1}{k^2} S_{11}. \tag{6}$$

The scattering ($C_{scat}$), absorption ($C_{abs}$), and extinction ($C_{ext}$) cross sections are related by the expression

$$C_{ext} = C_{scat} + C_{abs}. \tag{7}$$

The scattering efficiencies are given by the expression [17]

$$E_i = \frac{C_i}{G}. \tag{8}$$

$G$ is the particle cross-sectional area for $i$, being either extinction, absorption, or scattering.

The single-scattering albedo ($\omega_0$), which is a measure of the absorbing property of the scatterer, is given by

$$\omega_0 = \frac{C_{scat}}{C_{ext}}. \tag{9}$$

For $\omega_0 = 1$, the scatterer is nonabsorbing. The smaller the values of $\omega_0$ ($\omega_0 \ll 1$) imply that the scatterer is more absorbing. Generally, what the DDSCAT actually computes is the scattering coefficients of the particle. The relation between the scattering cross sections and coefficients is given by

$$Q_{coff} = C_{cross}(\lambda)\nu. \tag{10}$$

where $Q_{coff}$ is the scattering coefficient, $C_{cross}$ is the scattering cross section, and $\nu$ is the density of the particle in the medium.

Other important parameters to compute are the linear depolarization ratio $\chi$ and the lidar ratio $\varphi$. The linear depolarization ratio gives us an idea about the particle's ability to depolarize backscattered light. It is calculated by the expressions [6].

$$\chi = \frac{S_{11}(180) - S_{22}(180)}{S_{11}(180) + S_{22}(180)}. \tag{11}$$

The lidar ratio is given by

$$\varphi = \frac{C_{ext}}{C_{back}} = \frac{k^2 C_{ext}}{S_{11}(180)}, \tag{12}$$

where $S_{11}(180)$ is an indicator of the intensity of backscattered light.

### 2.2. Scattering Calculations in DDSCAT-7.3

To model the impacts of the ambient medium, size, surface roughness, and the imaginary part of the particle complex refractive index, the shape of the particle is fixed and assumed spherical. The medium refractive index is varied from dry air to water (1–1.33); the particle's dimensionless size parameter $x = 2\pi a_{eff}/\lambda$, where $a_{eff}$ is the particle's effective radius, is varied from 5 to 17.6; a surface roughness length is induced in the particle; and the imaginary part of the complex refractive index of the particle ($m = 1.48 + 0.0048i$) varies between 0.0015 and 0.1. The choice of $m$ has been published by Biagio et al. [26]. To investigate the impact of the incident wavelength, different light sources are used as incident light with a wavelength ranging from UV (0.3 µm) to infrared (1.4 µm) for the same spherical particle shape and complex refractive index. The particle shape impact is investigated by simulating five different Gaussian random spheres (GRS1, GRS2, GRS3, GRS4, and GRS5) using the SIRIS version 4 software. The shape file is then converted to a DDSCAT shape file using the nanoHUB online tool found at https://nanohub.org/login?return=L3Rvb2xzL2RkYWNvbnZlcnQvc2Vzc2lvbj9zZXNzPTIwMjk0ODM= (accessed on 20 April 2023). Since the particles are irregular and asymmetric, the orientation of the particle in space relative to the incident plane is a major factor that can affect aerosols scattering computation. For randomly oriented, irregular, asymmetric particles, we apply the method of orientational averaging since we cannot tell the orientation of the particle in space at a particular time. Three angles, $\beta$, $\Theta$, and $\Phi$, are used to describe the angular orientation of the particle in space. In this work, for efficient computation, we have set $\beta$ and $\Phi$ to run from $0 < (\beta, \Phi) < 360°$ with $N = 4$ intervals, which make use of the mid-point averaging rule, and $\Theta$ varies in the interval $0 < \Theta < 180°$ with $N = 3$ intervals, which make use of the Simpson's rule since $N$ is odd. We average the calculations over 48 angular orientations of the target and 2 incident polarizations.

To model the impact of the composition, we use the data published in Table 1 in the paper by H. Lindqvist et al. [6] on the complex refractive index of hematite and kaolinite. We assigned different amounts of dipoles to different complex refractive indices present in the composite material (% concentration of minerals), and we compared the scattering properties with pure kaolinite-like particles while fixing the shape and size of the particle. To investigate the impact of the single-particle mixing state, we simulate nine different locations of the impurity in the particle, including three lump positions, four surface incursions, and random and homogeneous mixing. A plane-polarized electric field (see Figure 1) with wavelength $\lambda = 0.5$ µm is used as an incident light source on the target particle except for the wavelength analysis. The number of dipoles ranges between $N = 90,000$ and $110,000$, fitting within the $48 \times 48 \times 48$ computational volume. The effective radius of the dipoles is $a_{eff} = 5 \times \lambda/2\pi = 0.4$ µm, and $|m|kd = 0.309$ is the choice of the dipole spacing where $m$ is the complex refractive index. We use the GPFAFT package that uses the conjugate gradient method without agonizing pain (GPFA) algorithm over the intel math kernel library (FFTMKL) as our choice for the fast Fourier transform algorithm since it is written in Fortran-90 and given that the intel math kernel library is not available

on our system. For the DDA method, we used the GKDLDR package, a modified lattice dispersion relation prescription of Gutkowicz-Krusin and Draine (2004). Since we are dealing with single particles with diameters in the range of microns, it is logical to compute the scattering field very close to the particle (near-field).

We calculate and compare the six Mueller matrix elements: scattering, absorption, and extinction efficiencies; single-scattering albedo; and the linear depolarization ratio for each comparative study.

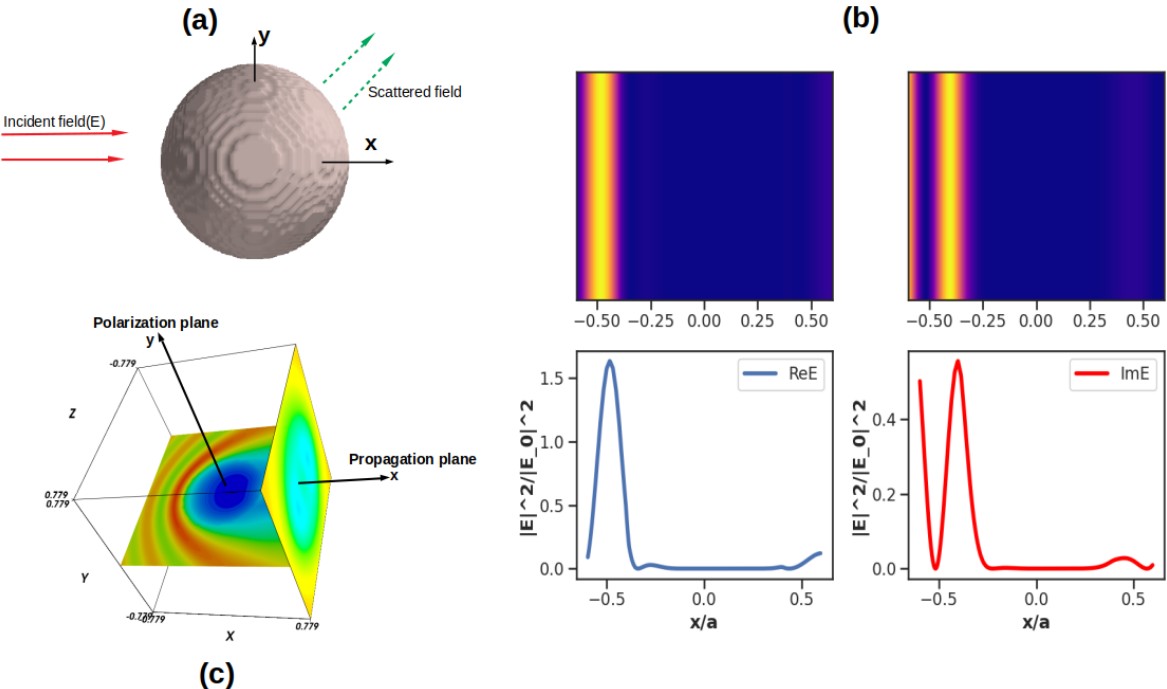

**Figure 1.** Shape of an aerosol dust particle (**a**) used to test for the impacts of the size, surface roughness, ambient medium, and the imaginary part of the complex refractive index. The real and imaginary parts of the plain-polarized incident electric field vector pass through the center of the particle (**b**), with $x/a$ being the particle diameter and $|E|^2/|E_0|^2$ the intensity, and (**c**) is the 3D diagram of the plain-polarized incident electric field showing the polarization and propagation plains. (**a**,**c**) were generated using the Mayavi2 open-source software

## 3. Results

### 3.1. Impact of the Refractive Index

The effect of an increase in the refractive index of the ambient medium surrounding the dust aerosol particle on its scattering, absorption, and extinction coefficients, which are proportional to their efficiencies, is shown in Figure 2. This figure was plotted for different values of the complex refractive index of the dust particle. Here, we see that all of the efficiencies decline with an increase in the refractive index of the ambient medium for the different values of the complex refractive index of the dust particle. However, the increase in the imaginary part of the complex refractive index results in an increase in light absorption by the particle, as shown in the absorption plot. Additionally, the single-scattering albedo, which is computed following Equation (13), is shown to be the near inverse of the absorption cross-section. This supports the finding that the increase in the imaginary parts of the complex refractive index results in an increase in light absorption via the aerosol particle. So, more absorbing particles will have a higher imaginary part of the complex refractive index.

Comparing the six relative Mueller matrix elements plotted against the scattering angle for aerosol particles in the air and in water, for the $m = 1.48 + 0.0048i$ complex refractive index of the dust aerosol, the simulated results presented in Figure 3 show

that the intensity of the scattered field reduces for particles in water compared to air. This can also be seen in Table 1, where all of the coefficients and albedo decrease from dry air to water. The polarization states also depend on the refractive index of the ambient medium. The optical properties are simulated by considering an incident wavelength of $\lambda = 0.5$ μm and assuming a spherically shaped aerosol particle. In Figure 4, the scattering properties of the dust aerosol particle in water relative to air are presented. This figure gives negative scattering matrices, signifying the scattering efficiency in the air is higher than in water. This figure equally exposes a strong symmetry between the diagonal Mueller matrix elements and between the off-diagonal elements for homogeneous and isotropic particles. From Figure 5 and Table 1, the imaginary part of the complex refractive index of the dust aerosol has been increased from $0.0048i$ to $0.1i$. Here, we see that the scattering intensities, scattering cross-sections, and albedo decrease with an increase in the imaginary part of the complex refractive index, signifying the importance of accurately estimating the imaginary part of the complex refractive index of aerosols for aerosol theoretical and experimental studies.

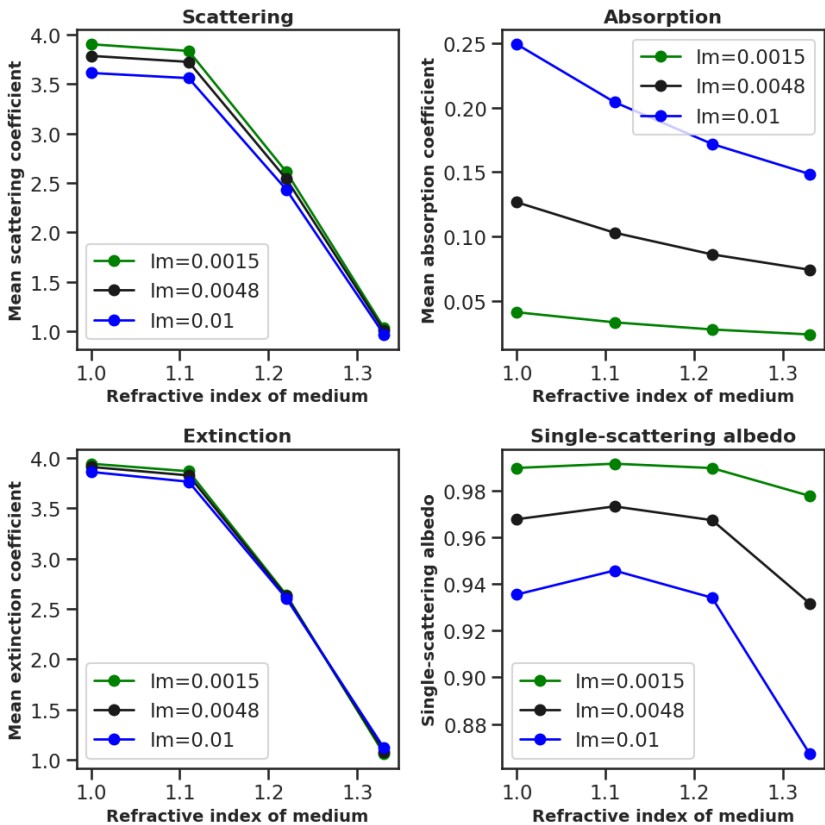

**Figure 2.** The relation between the mean scattering, absorption, extinction coefficients (efficiencies), single-scattering albedo, and the refractive index of the ambient medium. Graphs are plotted for different values of the complex refractive index of the dust aerosol. All of the coefficients are shown to decline with an increase in the refractive index of the ambient medium, but the increase in the imaginary part of the complex refractive index increases the light absorption by the particle.

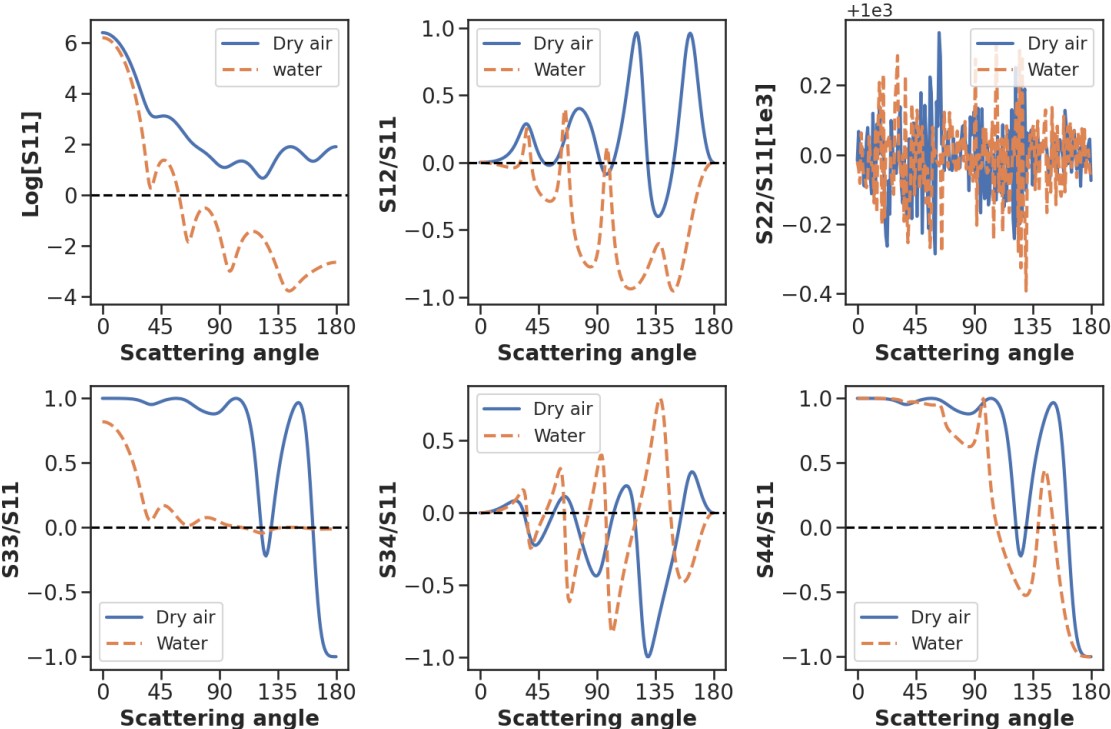

**Figure 3.** The effect of ambient medium on aerosol scattering properties. The refractive index of the medium varies from dry air to water (i.e, 1.0 to 1.33). A spherically shaped particle with a complex refractive index $m = 1.48 + 0.0048i$ and size parameter $x = 5$ at $\lambda = 0.5$ µm was used. This figure shows that the intensity of the back-scattered field reduces with an increase in the refractive index of the ambient medium. The polarization states also depend on the refractive index of the ambient medium. The optical properties are simulated by considering an incident wavelength of $\lambda = 0.5$ µm.

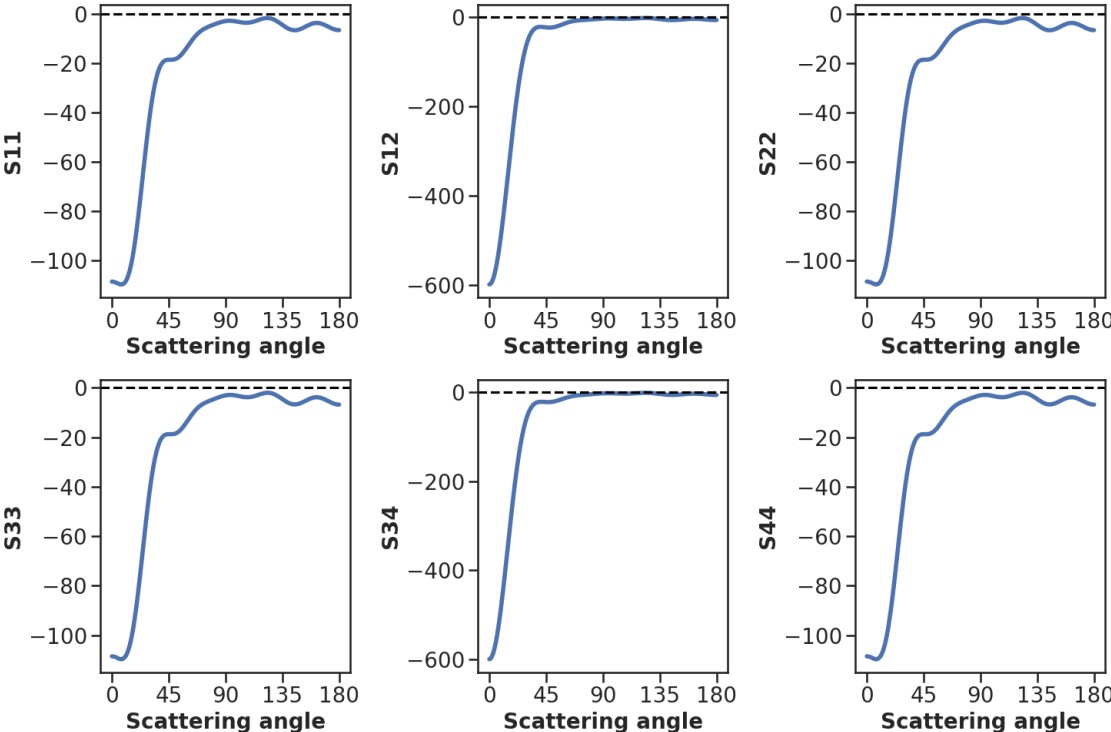

**Figure 4.** The scattering properties of the dust aerosol particle in water relative to dry air reveals symmetry ($S_{11} = S_{22}$, $S_{33} = S_{44}$ and $S_{12} = S_{21} = S_{34}$) in particle structure.

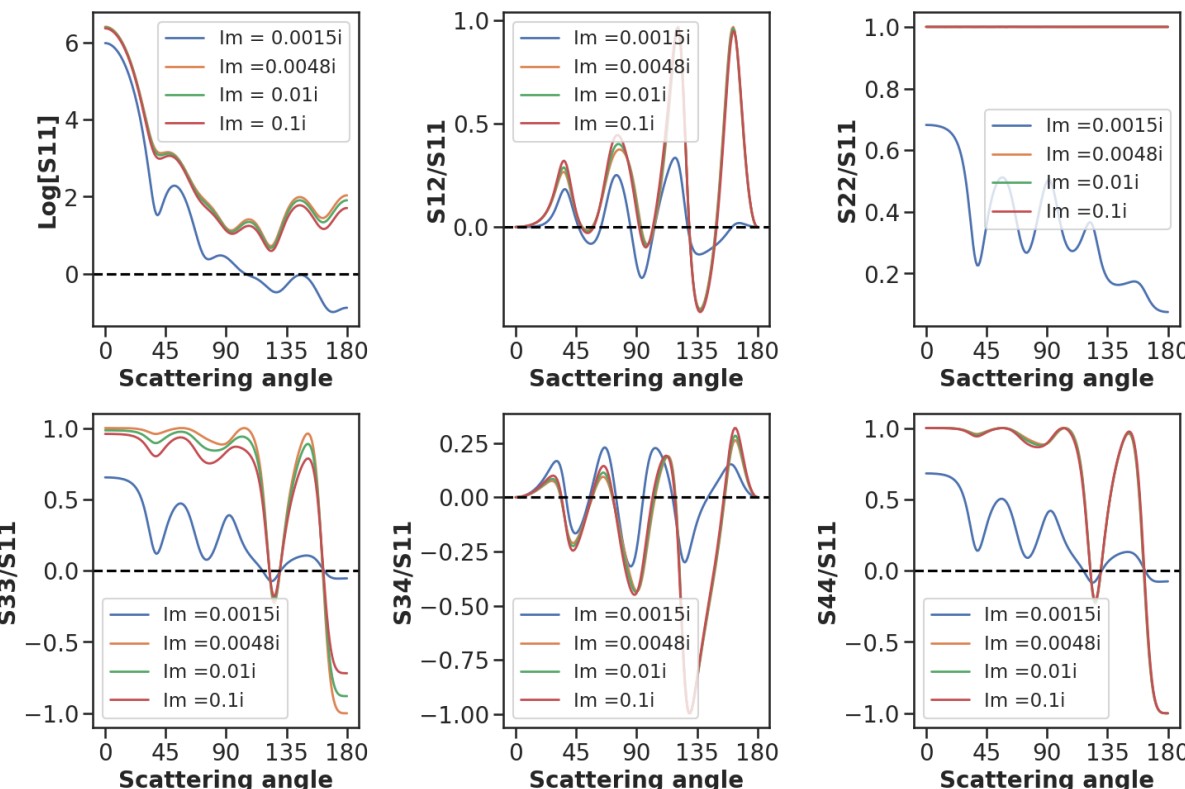

**Figure 5.** The effect of varying the imaginary part of the complex refractive index of the aerosol particle on its light-scattering properties. The ambient medium is considered to be dry air. Here, the real part of the complex refractive index of the dust aerosol is $m = 1.48$. This figure shows that the intensity of the back-scattered field reduces with an increase in the imaginary part of the complex refractive index of the aerosol. The polarization states are also shown to be dependent on the refractive index of the particle. The optical properties are simulated by considering an incident wavelength of $\lambda = 0.5\ \mu m$, assuming the particle is spherical with a size parameter $x = 5$.

### 3.2. Impact of the Incident Wavelength

To investigate the impact of the incident wavelength, we vary the wavelength of the incident light from the ultraviolet (UV) region of the electromagnetic spectrum to the infrared in both air and marine environments. A spherically shaped particle with a complex refractive index $m = 1.48 + 0.0048i$ and size parameter $x = 5$ at $\lambda = 0.5\ \mu m$ was used. From Figure 6, the relationship between the mean scattering, absorption, extinction coefficients/efficiencies, single-scattering albedo, and wavelength of the incident light is investigated. The middle region in pink is the visible region. Here, we see that the visible region has the highest scattering and extinction efficiency and single-scattering albedo (SSA) in both dry air and water compared to the UV and infrared regions. The plots equally show a higher curve for the water medium for absorption compared to air but lower curves for scattering, extinction, and albedos. Table 1 shows that particles under visible light have the highest scattering efficiencies and backscattering intensities, while for those under infrared, particles have the fewest backscattering intensities but high absorption power compared to particles under UV light. Hence, light in the visible region is more efficient for aerosol scattering experiments.

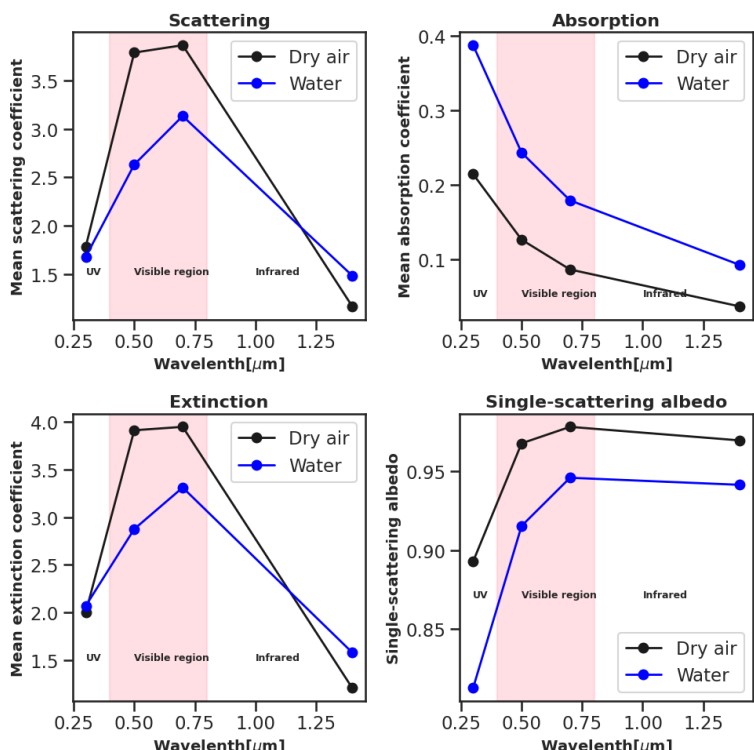

**Figure 6.** The relation between the mean scattering, absorption, extinction coefficients (efficiencies), single-scattering albedo, and wavelength of the incident light. A spherically shaped particle with a complex refractive index $m = 1.48 + 0.0048i$ and size parameter $x = 5$ at $\lambda = 0.5$ μm was used. Graphs are plotted for different values of the wavelength that ranges from the infrared to the ultraviolet region in both dry air and water. The middle region in pink is the visible region. Visible light has higher scattering and absorption efficiencies and SSA in both dry air and water compared to UV and infrared.

### 3.3. Impact of Size and Shape Distribution

To model the effect of particle size on its optical properties, we used two spherical dust particles with different effective radii (or size parameters) and the same complex refractive index ($m = 1.48 + 0.0048i$). A visible light source of wavelength 0.5 μm is incident on the particle in dry air. From Table 1, we see that particles with a diameter near the wavelength of the incident radiation have higher scattering, absorption, and extinction efficiencies compared to larger particles. The larger particle equally becomes less absorbing according to the single-scattering albedo values. Moreover, the value of $S_{11}(180)$ is shown to be slightly higher for larger particles than for smaller particles. Signifying smaller dust particles will increase the emission of thermal radiation when compared to larger particles but will lead to a reduction in the intensity of back-scattering.

To model the impact of the shape of aerosol particles on its optical properties, we simulate five different Gaussian random spheres with different aspect ratios (top row of Figure 7). From the literature, dominant aerosol shapes are either spherical or elliptic depending on the composition mixing [39]. So, we model five different shapes that range from elliptic to spherical. Since the particles shown in Figure 7 are irregular and asymmetric, we apply the method of orientational averaging, where we average the calculations over 48 angular orientations of the target and 2 incident polarizations. Figure 8 shows the different shapes have different scattering matrices, and, most importantly, the different shapes have distinct linear polarizations. Moreover, the intensity of the backscattering increases as the particle becomes less elongated and more spherical. Additionally, from Table 1, there is a decrease in the scattering coefficients as the particle becomes more

spherical ($\chi \to 0$). These particles are nonabsorbing ($\omega \approx 1$) since the imaginary part of the Kaolinite-like particles is zero.

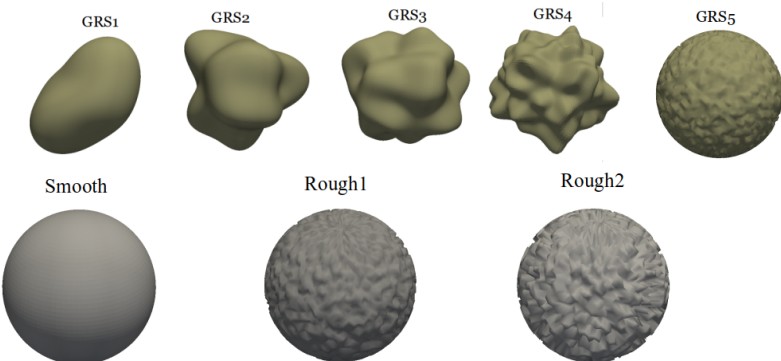

**Figure 7.** Model of the Gaussian randoms spheres (top row) to simulate the impact of shape on aerosol light-scattering properties. The models have different roughness lengths, radial standard deviations, and correlation angles. The model simulates light-scattering properties of different Kaolinite particles with complex refractive index $m_k = 1.562 + 0.00i$ at 0.5 μm wavelength of visible light and $x = 5$ size parameter. The bottom row (smooth, rough1, and rough2) are figures to simulate the impact of surface roughness. A surface roughness length is induced on a spherically shaped particle with a complex refractive index $m = 1.48 + 0.0048i$ and size parameter $x = 5$ at $\lambda = 0.5$ μm wavelength.

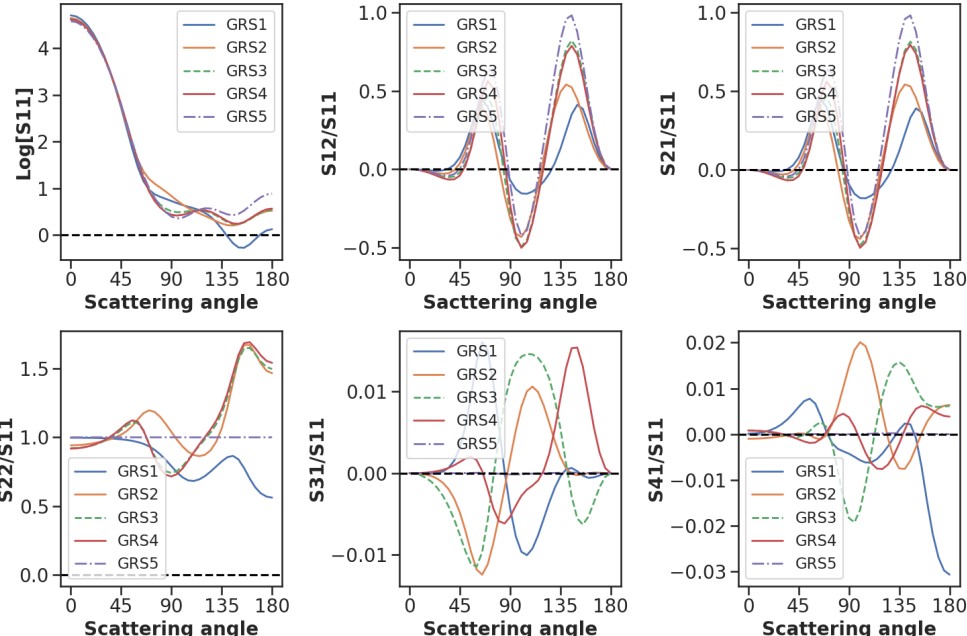

**Figure 8.** The impact of shape distributions on aerosol scattering matrices. Graphs are plotted for different Gaussian random spheres shown in the Figure 7 above. The method of orientational averaging has been used with the scattering computation averaged over 48 target orientations and 2 incident polarizations. Visible light of wavelength $\lambda = 0.5$ μm was used where the complex refractive index of Kaolinite is $m_k = 1.562 + 0.00i$ at $x = 5$ size parameter. Detail scattering results are presented in Table 1. This figure shows different shapes produce different scattering matrices and the intensity of the backscattering increases as the particle becomes less elongated and more spherical.

### 3.4. Impact of Surface Roughness

Surface roughness has been assumed to have a considerable impact on the light-scattering properties of aerosol but has not yet been experimentally investigated. In this section, we model the impact of surface roughness on aerosol optical properties by considering a spherical particle shape with different degrees of surface roughness as shown in the bottom row of Figure 7. Figure 9 shows that a small amount of induced surface roughness can significantly affect the degree of linear polarization of the particle but has a negligible effect on the scattering intensity. From Table 1, we see that all coefficients show no significant change when compared to the smooth particle, but there is a slight decrease in the intensity of the back-scattering as the roughness length increases. Figure 9 and the scattering results in Table 1, together with the results presented in Figure 8, show the impact of surface roughness is not as significant as the general particle structure but can affect the polarizability of the particle.

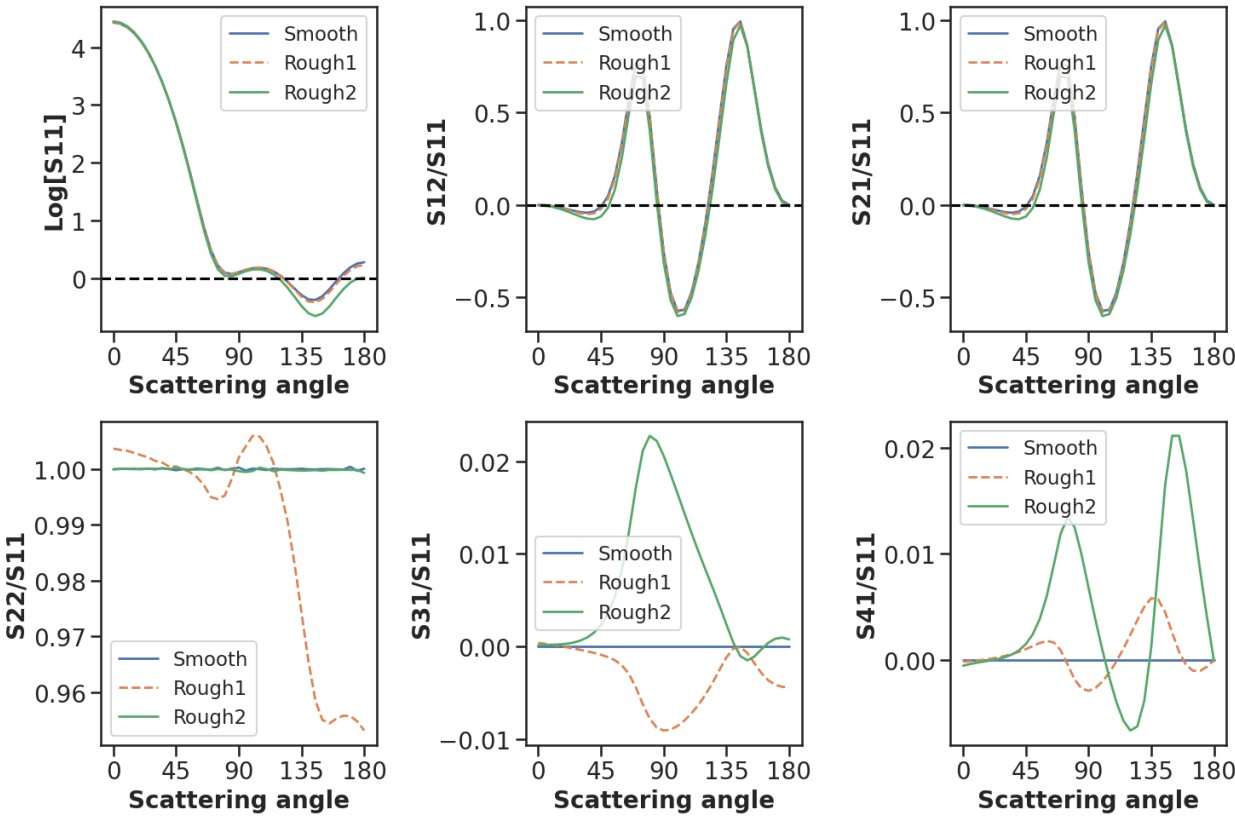

**Figure 9.** The impact of surface roughness on aerosol scattering properties. Graphs are plotted for the same particle shape (sphere) and size but different roughness lengths. The complex refractive index of the dust particle and the wavelength of the incident light have been fixed ($n = 1.48 + 0.0048i$ and 0.5 μm). Detail scattering results are presented in Table 1. This figure shows a small amount of surface roughness can significantly affect the degree of linear polarization of the particle but has a negligible effect on the scattering intensity as shown in Table 1.

### 3.5. Impact of Composition

To model the impact that composite dust particles have on aerosol light-scattering properties, we doped a kaolinite particle with a small amount of hematite. The percentage of kaolinite in the composite material is 98%, while hematite has a 2% concentration (Biagio et al. [40] have used 3–10% total iron in their study). Firstly, the pure kaolinite (100% concentration) optical properties are computed. Secondly, the kaolinite (98% concentration) is doped with hematite (2% concentration) with all dipoles localized as a lump. Thirdly, the 2% hematite on the surface; one randomly or homogeneous coating

and the other inhomogeneous coating; and, lastly, the 2% hematite are randomly and homogeneously distributed throughout the particle. In total, we simulate over nine different hematite distributions in the kaolinite particle. We compute the means and standard deviations of the asymmetric distributions of the impurities for both the lumps and surface incursions and compare their results with the pure particle, random, and homogeneous distribution of the impurity for three different incident wavelengths (380, 500, and 650 nm). Figure 10 is the tomography image when the particle in Figure 11 is sliced along the z-axis in the x-y plane to show the location of the kaolinite dipole particles (grey) and hematite impurity (dark red). While the particle shape and percentage of the impurity are the same, only the z-levels with the highest inclusion content for all models used are shown. Complete tomographic images can be found in Supplementary Figures S1–S9.

Since the particle shown in Figure 11a is irregular and asymmetric, we equally apply the method of orientational averaging, where we average the calculations over 48 angular orientations of the target and 2 incident polarization. The calculations were carried out for three incident wavelengths (380, 500, and 650 nm). Figure 11a shows the single aerosol-particle sample shape (kaolinite) and spectrum, obtained via the scanning electron microscope (SEM) of the Saharan mineral dust sample with a small percentage of iron impurity, and (b) is the particle shape model used (GRS3) to mimic the shape in the image. The complex refractive index of Kaolinite is taken as $m_k = 1.562 + 0.00i$, while that of Hematite is $m_h = 3.1020 + 0.0925i$ at $\lambda = 550$ nm [6]. Figure 12 shows the impacts of the various composition's mixing states (pure, random/homogeneous, lump, and surface models) on the relative scattering of Mueller matrix elements for 500 nm incident wavelength. This figure results in Table 2 under all incident wavelengths used in this work, which show the existence of impurities as lumps in the particle can significantly alter the intensity and polarization states compared to pure, random, and surface mixing. We equally found that both the random and homogeneous mixing produce the same scattering matrices. Hence, for simulation and experimental purposes, anyone can be used. Additionally, the variation in the scattering intensities for pure, homogeneous, random, lump, and surface models can vary up to a factor of 0.1, signifying the impact of the mixing state of the aerosol compositions on its optical properties. The scattering matrices for the other incident wavelengths (380 and 650 nm) can be found in Supplementary Figures S10 and S11. From Supplementary Table S1, comparing the scattering properties of the different lumps models at different incident wavelengths, we find that the light-scattering properties of aerosols depend strongly on the location of the lump impurities in the sample. When the lump is located near the center of the particle (see Supplementary Figure S5), the particle exhibits the least scattering efficiencies compared to the other lump models (peripheral locations) (see Supplementary Figures S3 and S4) except for the 650 nm incident wavelength. Additionally, when the impurity is uniformly distributed over the particle's surface, the scattering efficiency is the least compared to the other asymmetric surface distributions. This analysis also shows that the scattering properties of composite aerosol particles strongly depend on the mixing state of the impurities in the sample, but the presence of a small number of impurities will make the composite particle more absorbing compared to the pure particle.

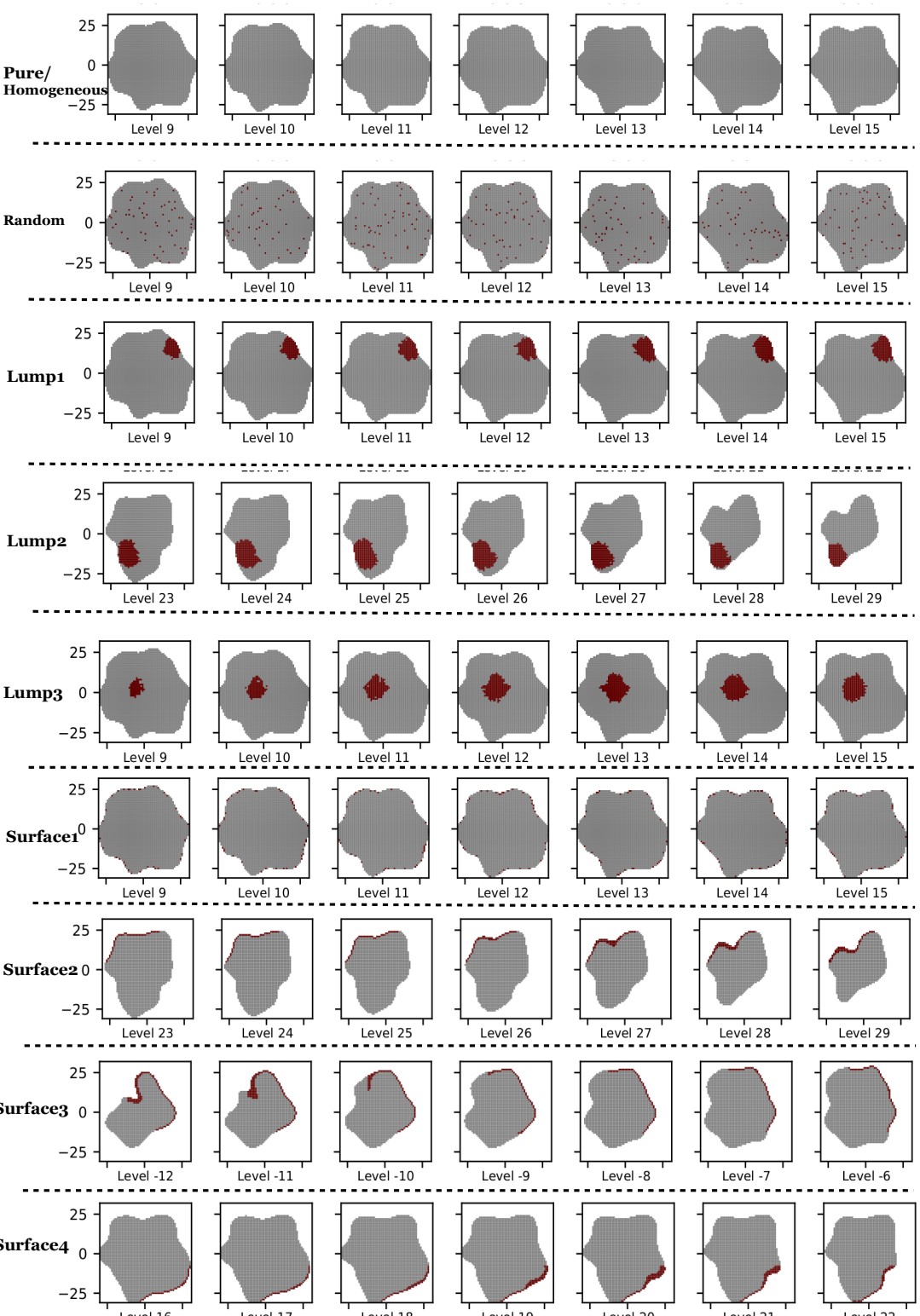

**Figure 10.** Tomography images when the particle in Figure 11b is sliced along the z-axis in the x-y plane to show the location of the pure dipole particles (grey) and impurities (dark red). All models (surface and internal incursion) used in this study are shown above. While the particle shape is the same for all impurities, only the z-levels with the highest inclusion content are shown. Complete tomographic images can be found in Supplementary Figures S1–S9, respectively.

**(a)**

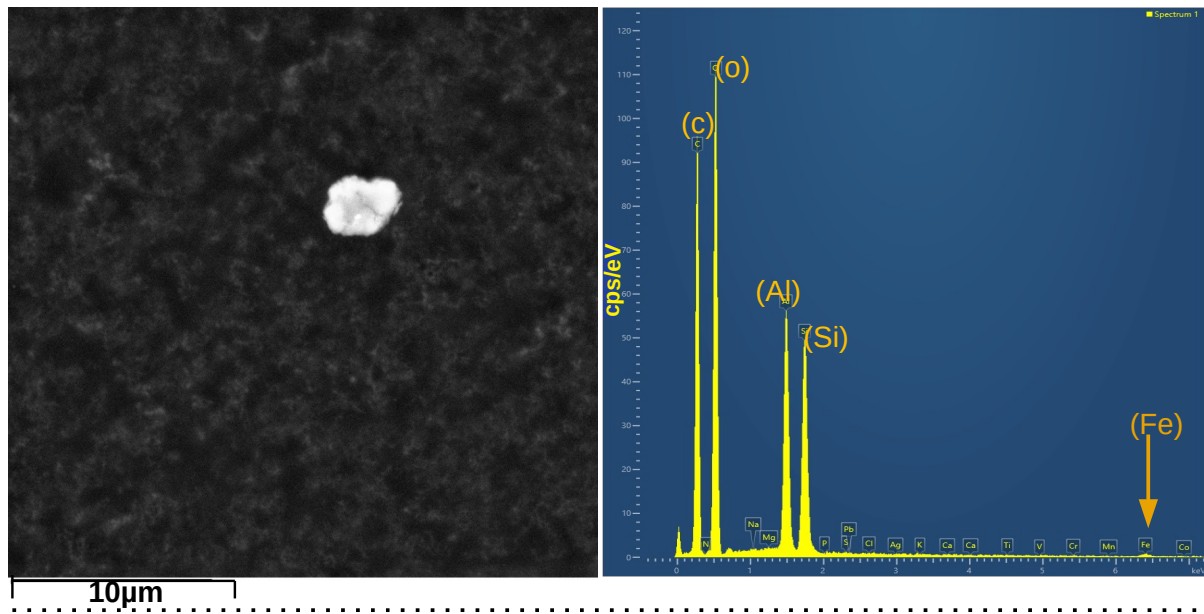

**(b)**

**GRS3**

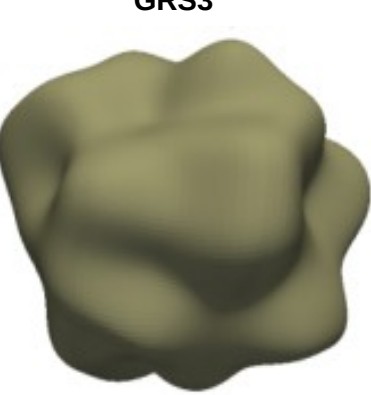

**Figure 11.** (**a**) Single aerosol-particle sample shapes (kaolinite) and their spectra, obtained via the scanning electron microscope (SEM) of Saharan mineral dust sample with a small percentage of iron impurity, and (**b**) is the particle shape model used (GRS3) to mimic the shape in the SEM image. The particle is then dopped with 98% kaolinite and 2% hematite at different mixing states. The complex refractive index of kaolinite is taken as $m_k = 1.562 + 0.00i$, while that of Hematite is $m_h = 3.1020 + 0.0925i$. The scattering results are listed in Table 1 for comparison purposes.

**Table 1.** Optical parameters summary table and sensitivity analysis for different external conditions and conditions affecting the particle as a whole.

| Parameters | Models | $\lambda$ (nm) | x | Medium | RI | $Q_{ext}$ | $Q_{abs}$ | $Q_{scat}$ | $\omega$ | $S_{11}(180)$ | $S_{22}(180)$ | $\chi$ |
|---|---|---|---|---|---|---|---|---|---|---|---|---|
| Shape | GRS1 | 500 | 5 | air | $1.562 + 0.00i$ | 3.967 | $2.46 \times 10^{-19}$ | 3.967 | $\sim 1$ | 1.128 | 0.63 | 0.028 |
| | GRS2 | 500 | 5 | air | $1.562 + 0.00i$ | 3.912 | $-2.0 \times 10^{-19}$ | 3.9137 | $\sim 1$ | 1.678 | 1.654 | 0.0072 |
| | GRS3 | 500 | 5 | air | $1.562 + 0.00i$ | 3.852 | $5.577 \times 10^{-21}$ | 3.852 | $\sim 1$ | 1.709 | 1.689 | 0.0059 |
| | GRS4 | 500 | 5 | air | $1.562 + 0.00i$ | 3.853 | $6.02 \times 10^{-19}$ | 3.853 | $\sim 1$ | 1.755 | 1.739 | 0.0046 |
| | GRS5 | 500 | 5 | air | $1.562 + 0.00i$ | 3.8067 | $2.462 \times 10^{-19}$ | 3.8068 | $\sim 1$ | 2.416 | 2.416 | 0.00 |
| Size parameter (x) | Sphere | 500 | 5.0 | air | $1.48 + 0.0048i$ | 3.192 | 1.176 | 2.016 | 0.632 | 0.414 | 0.414 | 0.0 |
| | | 500 | 17.6 | air | $1.48 + 0.0048i$ | 2.301 | 1.132 | 1.169 | 0.51 | 2.247 | 2.247 | 0.0 |
| Surface | Smooth | 500 | 5 | air | $1.48 + 0.0048i$ | 3.2995 | 0.0677 | 3.2317 | 0.979 | 1.3209 | 1.321 | $\sim 0.0$ |
| roughness | Rough1 | 500 | 5 | air | $1.48 + 0.0048i$ | 3.2995 | 0.0676 | 3.2319 | 0.979 | 1.2586 | 1.259 | $\sim 0.0$ |
| | Rough2 | 500 | 5 | air | $1.48 + 0.0048i$ | 3.2919 | 0.0672 | 3.2247 | 0.979 | 1.0137 | 1.013 | $\sim 0.0$ |
| Wavelength | Sphere | 300 | 5 | air | $1.48 + 0.0048i$ | 2.0 | 0.214 | 1.78 | 0.89 | 5.33 | 5.33 | 0.0 |
| | Sphere | 500 | 5 | air | $1.48 + 0.0048i$ | 3.91 | 0.12 | 3.78 | 0.967 | 6.73 | 6.73 | 0.0 |
| | Sphere | 700 | 5 | air | $1.48 + 0.0048i$ | 3.95 | 0.086 | 3.86 | 0.977 | 1.96 | 1.96 | 0.0 |
| | Sphere | 1400 | 5 | air | $1.48 + 0.0048i$ | 1.21 | 0.037 | 1.17 | 0.967 | 0.04 | 0.04 | 0.0 |
| Ambient | Sphere | 500 | 5 | air | $1.48 + 0.0048i$ | 3.91 | 0.13 | 3.78 | 0.967 | 6.74 | 6.74 | 0.0 |
| | Sphere | 500 | 5 | water | $1.48 + 0.0048i$ | 1.08 | 0.074 | 1.01 | 0.93 | 0.0713 | 0.0713 | 0.0 |
| Particle | Sphere | 500 | 5 | air | $1.48 + 0.0015i$ | 3.943 | 0.041 | 3.902 | 0.989 | 7.658 | 7.658 | 0.0 |
| Im (RI) | Sphere | 500 | 5 | air | $1.48 + 0.0048i$ | 3.911 | 0.13 | 3.784 | 0.967 | 6.738 | 6.738 | 0.0 |
| | Sphere | 500 | 5 | air | $1.48 + 0.01i$ | 3.862 | 0.25 | 3.612 | 0.935 | 5.508 | 5.508 | 0.0 |
| | Sphere | 500 | 5 | air | $1.48 + 0.1i$ | 3.192 | 1.176 | 2.016 | 0.631 | 0.414 | 0.414 | 0.0 |

$Q_{ext}$, $Q_{abs}$, and $Q_{scat}$ are the extinction, absorption, and scattering coefficients, respectively. $\omega$ and $\chi$ are the single-scattering albedo and the linear depolarization ratio, respectively, while $S_{ii}$ are the Mueller matrix elements for intensities.

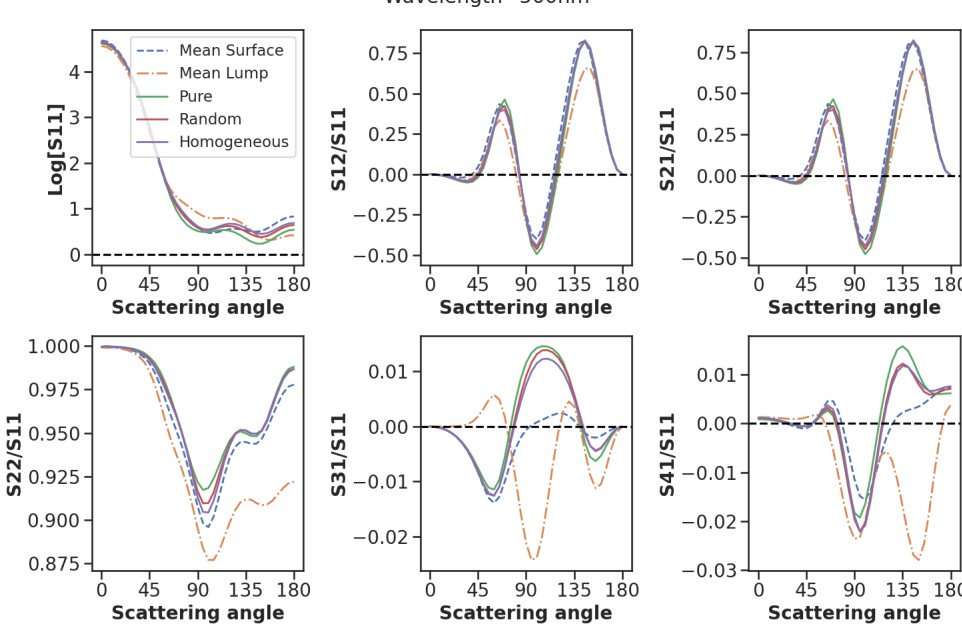

**Figure 12.** Simulated single-scattering Mueller matrix elements for the aerosol particle models shown in Figure 10 with different chemical compositions and mixing states. The averages of the lumps and surface incursion models have been plotted and compared with other models. The method of orientational averaging has been used with the scattering computation averaged over 48 target orientations and 2 incident polarizations. The plot for visible light of wavelength $\lambda = 500$ nm has been used here where the complex refractive index of Kaolinite is $m_k = 1.562 + 0.00i$, while that of Hematite is $m_h = 3.1020 + 0.0925i$. Comparing these plots with those for incident wavelengths of 380 and 650 nm, found in the Supplementary Figures S10 and S11, we see that the light-scattering properties of single-particle aerosols are strongly wavelength dependent as their scattering matrices are completely different. This figure equally shows that the light-scattering properties of aerosols can be significantly enhanced just by introducing a small amount of impurity at different mixing states in the compound.

**Table 2.** Optical parameters summary table and sensitivity analysis for the impact of composition at different wavelengths. The GRS3 shape model was used for the x = 5 size parameter, where the complex refractive index of Kaolinite is $m_k = 1.562 + 0.00i$ and that of Hematite is $m_h = 3.1020 + 0.0925i$ with 98% and 2% concentrations, respectively, in dry air condition.

| Parameters | Models | $Q_{ext}$ | $Q_{abs}$ | $Q_{scat}$ | $\omega$ | $S_{11}$ (180) | $S_{22}$ (180) | $\chi$ |
|---|---|---|---|---|---|---|---|---|
| Composition ($\lambda$ = 380 nm) | Pure kaolinite | 4.239 | $-5.95 \times 10^{-19}$ | 4.239 | 1 | 4.118 | 3.95 | 0.021 |
| | Random | 4.228 | 0.019 | 4.209 | 0.995 | 4.85 | 4.635 | 0.023 |
| | Homogeneous | 4,23 | 0.023 | 4.21 | 0.995 | 5.154 | 4.924 | 0.023 |
| | Surface (mean ± std) | 4.343 (0.015) | 0.052 (0.005) | 4.289 (0.012) | 0.987 (0.833) | 6.667 (0.001) | 6.327 (0.783) | 0.026 (0.007) |
| | Lump (mean ± std) | 3.962 (0.160) | 0.078 (0.011) | 3.885 (0.168) | 0.980 (0.003) | 3.771 (0.407) | 2.796 (0.555) | 0.015 (0.117) |
| Composition ($\lambda$ = 500 nm) | Pure kaolinite | 3.852 | $5.577 \times 10^{-21}$ | 3.852 | 1 | 1.709 | 1.689 | 0.0059 |
| | Random | 3.956 | 0.012 | 3.944 | 0.997 | 1.895 | 1.87 | 0.0053 |
| | Homogeneous | 4.01 | 0.015 | 3.997 | 0.997 | 1.98 | 1.95 | 0.0076 |
| | Surface (mean ± std) | 4.033 (0.006) | 0.031 (0.002) | 3.998 (0.002) | 0.991 (0.001) | 2.29 (0.168) | 2.239 (0.161) | 0.011 (0.002) |
| | Lump (mean ± std) | 3.838 (0.277) | 0.062 (0.008) | 3.775 (0.279) | 0.983 (0.003) | 1.517 (0.223) | 1.398 (0.166) | 0.039 (0.027) |
| Composition ($\lambda$ = 650 nm) | Pure kaolinite | 2.768 | $1.288 \times 10^{-20}$ | 2.768 | 1 | 0.77 | 0.763 | 0.00022 |
| | Random | 2.934 | 0.0086 | 2.926 | 0.997 | 0.752 | 0.75 | 0.0027 |
| | Homogeneous | 3.02 | 0.011 | 3.01 | 0.997 | 0.75 | 0.75 | 0.0033 |
| | Surface (mean ± std) | 2.977 (0.006) | 0.021 (0.002) | 2.957 (0.006) | 0.993 (0.0) | 0.662 (0.009) | 0.654 (0.011) | 0.0065 (0.001) |
| | Lump (mean ± std) | 3.021 (0.033) | 0.027 (0.007) | 2.993 (0.026) | 0.991 (0.002) | 0.67 (0.18) | 0.66 (0.178) | 0.0069 (0.004) |

$Q_{ext}$, $Q_{abs}$, and $Q_{scat}$ are the extinction, asorption and scattering coefficients, respectively. $\omega$ and $\chi$ are the single-scattering albedo and the linear deporarization ratio, respectively, while $S_{ii}$ are the Mueller matrix elements for intensities.

## 4. Discussion

Aerosol shape distribution is a major sensitive parameter that has been found to significantly affect aerosols' radiative properties. In this work, we have shown that the shape distribution of atmospheric aerosols has a significant effect on its linear polarizability, a similar finding that has been carried out by Li et al. [41] when they simulated the effects of the shape distribution of aerosol particles on their volumetric scattering properties and radiative transfer. Moreover, a study by Ladislav et al. [42] also shows that different aerosol shape distributions have a significant effect on night skyglow, pointing to the strong impact of size distribution on aerosol light-polarization. We also found that using the spherical shape model over the elliptic shape model can underestimate the scattering efficiencies by up to 4% for a pure kaolinite particle. A similar finding was by Mishra et al. [43] when they showed that the spherical shape model can underestimate single-scattering Albedo, SSA for the dust with low 1.1% hematite by 3.5%. Huang et al. [5] investigated the impact of the refractive index on the aerosol optical properties. Their results show that different RIs significantly affect the scattering matrices, but the experiment was carried out for the variation of the real part only. Here, we have shown that the more photosensitive imaginary part of the complex refractive index can affect the error in the estimation of the particle's single scattering albedo by up to 35.8%. Moreover, the different imaginary RIs significantly affect both the scattering intensities and the polarization of the particle. The choice of the imaginary RI also strongly depends on the wavelength of the incident radiation, shape, and composition of the particle. This makes the accurate estimation of the imaginary RI even more difficult.

The absorbing ability of aerosols has equally been found to depend on the incident wavelength and the composite minerals present in the particle. Hematite in the dust particle will absorb light differently in shortwave and longwave limits [4]. We have fixed the particle size, used a single mineral (iron) in the particle, and investigated the scattering properties when the particle is illuminated with different light sources, ranging from UV to infrared. Our findings show that the absorbing ability of iron composition declines with an increase in wavelength. Moreover, because different minerals in a particle have different refractive indices, the different scattering properties, as well as the same particle shape, size, and wavelength but with different inclusions, scatter light differently, pointing to more complex relations between particle shape, size, composition, RI, and wavelength for estimating the radiative properties of aerosols. In addition, we have also shown that not only do the different compositions in the particle affect its optical properties, but the way

the minerals are mixed is also another major factor provided their percent concentrations are kept constant.

## 5. Summary and Conclusions

The aerosol photosensitive parameters affect the aerosols' scattering properties in a complex way, which can affect the uncertainty in climate and environmental modeling studies. The impact of two or more parameters can give rise to new and unique properties. The contribution of each parameter in the presence of other parameters has not yet been studied. In this work, we have isolated and investigated the impacts of each parameter while keeping the others constant. By using the method of discrete-dipole approximation in DDSCAT, we have computed their mean scattering, absorption, and extinction coefficients for different angular orientations of the target. We have also computed and compared their single-scattering albedo, asymmetric parameters, and backscattering intensities. We have generated optical parameter tables for sensitivity analysis that can be used as a reference for any aerosol optical sensitivity study. We have found that water as a medium will lead to a decrease in the single-scattering albedo, scattering, and extinction efficiencies but increases the absorption of light. These findings tie in with the theory on the electromagnetic scattering of light in denser and less dense media and have also been shown in the paper [6]. Surface roughness has been found to strongly alter the polarizability of the aerosol particle but has a negligible effect on its single-scattering albedo. We equally compared the scattering properties of smaller and larger particles where we have seen that, for particles with a diameter comparable to the wavelength of the incident light, there is a higher scattering efficiency compared to particles with a diameter larger than the wavelength of incident light. Moreover, smaller particles show lower backscattering intensity than larger particles. These findings may be particularly useful for lidar remote sensing experiments. Additionally, we have seen that the intensity of the backscattering increases and scattering efficiency decreases as the particle becomes less elongated and more spherical. We have shown that using the spherical shape model over the elliptic shape model for aerosols can underestimate the scattering efficiencies by up to 4% for a pure kaolinite particle, and a similar finding has been shown by Mishra et al. [43] when they carried out a sensitivity analysis at 550 nm wavelength for dust with low 1.1% hematite. An increase in the imaginary part of the complex refractive index has been shown to decrease the intensity of backscattering and increase light absorption by the particle, and it can equally affect the error in the estimation of the particle's albedo by up to 35.8%. This shows the importance of accurately estimating the imaginary part of the complex refractive index of aerosols during experimental or theoretical studies.

The impact of composition shows that the difference in the scattering efficiencies between the pure kaolinite and the composite (kaolinite/hematite) when the impurity is randomly and homogeneously mixed can vary up to a factor of 0.1, while for the case of the impurity being localized as a lump, the light-scattering efficiencies of aerosols strongly depend on the location of the lump impurities in the sample. Moreover, the lump impurity model has been shown to have the lowest scattering efficiency and single-scattering albedo on average compared to all other models under study. Hence, the light-scattering efficiency of aerosols strongly depends on the single-particle mixing state of the aerosol compositions. Additionally, we have seen that the presence of a small number of impurities will make the composite particle more absorbant compared to the pure particle; a similar finding by [44] showed that the SSA of the polluted dust system is larger if polluted dust is considered as pure dust spheroid (with 4% hematite). The results in this paper can act as a reference for aerosol single-scattering experiments and computations.

**Supplementary Materials:** The following supporting information can be downloaded at: www.mdpi.com/xxx/s1, Table S1: Parameter summary table and sensitivity analysis; Figure S1: Tomography images when the particle in Figure 10b is sliced along the z-axis in the x-y plane for the pure kaolinite particle (Pure); Figure S2: Tomography images when the particle in Figure 10b is sliced along the z-axis in the x-y plane to show the location of the pure dipole particles (grey) and impurities (dark red) for the random model distribution of impurities (Random); Figure S3: Tomography images when the particle in Figure 10b is sliced along the z-axis in the x-y plane to show the location of the pure dipole particles (grey) and impurities (dark red). Here, the lump is located at the topmost surface of the particle (Lump1); Figure S4: Tomography images when the particle in Figure 10b is sliced along the z-axis in the x-y plane to show the location of the pure dipole particles (grey) and impurities (dark red). Here, the lump is located at the bottommost surface of the particle (Lump2); Figure S5: Tomography images when the particle in Figure 10b are sliced along the z-axis in the x-y plane to show the location of the pure dipole particles (grey) and impurities (dark red). Here, the lump is located near the center of the particle (Lump3); Figure S6: Tomography images when the particle in Figure 10b is sliced along the z-axis in the x-y plane to show the location of the pure dipole particles (grey) and impu-rities (dark red). Here, the uniformly distributed surface incursion is shown; Figure S7: Tomography images when the particle in Figure 10b is sliced along the z-axis in the x-y plane to show the location of the pure dipole particles (grey) and impurities (dark red). Here, the asymmetric distributed surface incursion is shown; Figure S8: Tomography images when the particle in Figure 10b is sliced along the z-axis in the x-y plane to show the location of the pure dipole particles (grey) and impurities (dark red). Here, the asymmetric distributed surface incursion is shown; Figure S9: Tomography images when the particle in Figure 10 is sliced along the z-axis in the x-y plane to show the location of the pure dipole particles (grey) and impurities (dark red). Here, the asymmetric distributed surface incursion is shown; Figure S10: Simulated single-scattering Mueller matrix elements for the aerosol particle models shown in Figure 11 with different chemical composi-tions and mixing states. The averages of the lumps and surface incursion models have been plotted and compared with other models. The method of Orientational Averaging has been used with the scattering computation averaged over 48 target orientations and 2 incident polarizations. The plot for visible light of wavelength $\lambda$ = 380 nm has been used here where the complex refractive index of Kaolinite is $m_k$ = 1.562 + 0.00$i$ while that of Hematite is $m_h$ = 3.1020 + 0.0925$i$; Figure S11: Simulated single-scattering Mueller matrix elements for the aerosol particle models shown in Figure 11 with different chemical compositions and mixing states. The averages of the lumps and surface incursion models have been plotted and compared with other models. The method of Orientational Averaging has been used with the scattering computation averaged over 48 target orientations and 2 incident polarizations. The plot for visible light of wavelength $\lambda$ = 650 nm has been used here where the complex refractive index of Kaolinite is $m_k$ = 1.562 + 0.00$i$ while that of Hematite is $m_h$ = 3.1020 + 0.0925$i$.

**Author Contributions:** Conceptualization, N.A.A. and K.K.; Methodology, N.A.A., K.K. and A.S.; Validation, N.A.A.; Investigation, K.K.; Resources, N.A.A., K.K. and A.S.; Writing—original draft, N.A.A.; Writing—review & editing, N.A.A., K.K. and A.S.; Visualization, N.A.A. and A.S.; Supervision, K.K. All authors have read and agreed to the published version of the manuscript.

**Funding:** This research received no external funding.

**Data Availability Statement:** See Supplementary Materials for supporting content.

**Acknowledgments:** The authors will like to thank B. T. Draine and P. J. Flatau for developing the DDSCAT and making it publicly available. This work has received financial support from the Alexander von Humboldt Foundation (AvH).

**Conflicts of Interest:** The authors, to the best of their knowledge, declare no competing interest.

**Sample Availability:** The implementation of the DDSCAT-7.3.3 model is available as open source on http://ddscat.wikidot.com/start (accessed on 20 April 2023).

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
