# Peer review of "Near-Field Single-Scattering Calculations of Aerosols: Sensitivity Studies"

_optics, doi:10.3390/opt4020028_

Round 1

Reviewer 1 Report

This paper is devoted to modeling the effects of the photosensitive parameters of aerosols on their optical properties. The paper is well-structured and make a good presentation of results. In my opinion, the paper could be published, but after addressing a few issues.

-          Line 171 – “size parameter” is first mentioned in the text but not defined. Please, provide the formula

-          Line 326 – “results in table ??” seems like the reference is broken

-          Please increase the label for the size bar as well as the Y-axis labels on the Figure 10

-          In the introduction to the manuscript authors describe the importance of the investigation of light scattering on the aerosol particles and points out the negative impact of this process on the climate change, lidar and satellite imaging. But the authors miss a few applications that becomes really relevant nowadays and that also suffer from the aerosol scattering. Just to name a few of them — free space optical communications, wireless energy transfer through the scattering and turbulent atmosphere, imaging objects through scattering atmospheric aerosol, wireless charging and/or removing unnamed aerial vehicles, space debris removal, etc. The results presented by the authors can be useful for researchers in these fields.
I would recommend to have a look at the following papers and add a few comments on it to the introduction.

·         Y. Shuto, “High-Power Laser Beam Propagation in Slightly Wet Atmosphere”, Journal of Electrical and Electronic Engineering 10(6), pp. 215-222, 2022. DOI:

·         10.11648/j.jeee.20221006.11 

·         I. Galaktionov, A. Kudryashov, J. Sheldakova, A. Nikitin, V. Samarkin, "Laser beam focusing through the atmosphere aerosol", Proc. SPIE 10410, pp. 104100M, 2017. https://doi.org/10.1117/12.2276180

-          There are grammatical and punctuational errors (though their number is really small), i.e., in line 119 the comma “,” after “section 2” has to be removed; in line 329 comma “,” in “homogeneous mixing, produce” has to be removed, etc.

Author Response

Response to Reviewer 1

This paper is devoted to modeling the effects of the photosensitive parameters of aerosols on their optical properties. The paper is well-structured and make a good presentation of results. In my opinion, the paper could be published, but after addressing a few issues.

Response: Thank you very much for your complements and recommendations. I have addressed the issues you have raised accordingly.

-Line 171 – “size parameter” is first mentioned in the text but not defined. Please, provide the formula

Response: This has been resolved. Find the correction underlined in blue.

-Line 326 – “results in table ??” seems like the reference is broken

Response: This has been resolved. Find the correction underlined in blue.

-Please increase the label for the size bar as well as the Y-axis labels on the Figure 10

Response: This has been resolved. Find the verify to confirm in the new plot

.

-In the introduction to the manuscript authors describe the importance of the investigation of light scattering on the aerosol particles and points out the negative impact of this process on the climate change, lidar and satellite imaging. But the authors miss a few applications that becomes really relevant nowadays and that also suffer from the aerosol scattering. Just to name a few of them — free space optical communications, wireless energy transfer through the scattering and turbulent atmosphere, imaging objects through scattering atmospheric aerosol, wireless charging and/or removing unnamed aerial vehicles, space debris removal, etc. The results presented by the authors can be useful for researchers in these fields.
I would recommend to have a look at the following papers and add a few comments on it to the introduction.

·Y. Shuto, “High-Power Laser Beam Propagation in Slightly Wet Atmosphere”, Journal of Electrical and Electronic Engineering 10(6), pp. 215-222, 2022. DOI:10.11648/j.jeee.20221006.11

·I. Galaktionov, A. Kudryashov, J. Sheldakova, A. Nikitin, V. Samarkin, "Laser beam focusing through the atmosphere aerosol", Proc. SPIE 10410, pp. 104100M, 2017. https://doi.org/10.1117/12.2276180

Response: I have added some more applications of the scattering phenomenon including the those described in the papers you recommended.

-There are grammatical and punctuational errors (though their number is really small), i.e., in line 119 the comma “,” after “section 2” has to be removed; in line 329 comma “,” in “homogeneous mixing, produce” has to be removed, etc.

Response: Some cross-checking has been done and the punctuation errors highlighted here have been corrected.

Thank you ones more for your significant contribution to improve the quality of our paper.

Reviewer 2 Report

Dear editor, by reading through the paper, I think this manuscript has given a novelty idea on studying the physicochemical property of aerosol by measuring their photosensitive parameters. The idea is similar with particle size measurement by optical method, however, this study also consider further influence factors. This paper contains sufficient literature investigation and experimental results, thus on general it is suitable for publication on optics after consider the following suggestions:

1. Line 84: The full name of DDSCAT should be shown here.

2. In Figure 1, the author should clarify the legend meaning in the caption. E.g. x/a to make it easier for the reader to understand.

3. Line 171: The particle’s 170 size parameter is varied from 5 to 17.6 μm?

For table 1 and table 2, I recommend the authors add a footnote to show detail information of Qabs, Qscat and so on.

4. Line 326: What is Table ?? meaning for?

5. The authors can add some discussion on the potential application prospect on environmental management and monitoring.

Author Response

Response to Reviewer 2

Dear editor, by reading through the paper, I think this manuscript has given a novelty idea on studying the physicochemical property of aerosol by measuring their photosensitive parameters. The idea is similar with particle size measurement by optical method, however, this study also consider further influence factors. This paper contains sufficient literature investigation and experimental results, thus on general it is suitable for publication on optics after consider the following suggestions:

Response: Thank you very much for your complements and recommendations. I have addressed the issues you have raised accordingly.

1. Line 84: The full name of DDSCAT should be shown here.

Response: This has been resolved. Find the correction underlined in blue.

2. In Figure 1, the author should clarify the legend meaning in the caption. E.g. x/a to make it easier for the reader to understand.

Response: This has been resolved. Find the correction underlined in blue.

3. Line 171: The particle’s 170 size parameter is varied from 5 to 17.6 μm?

Response: The size parameter is dimensionless. The statement line has been modified to include the word ‘dimensionless’ for clarity.

For table 1 and table 2, I recommend the authors add a footnote to show detail information of Qabs, Qscat and so on.

Response: This has been resolved. Footnotes have been added. Thank you.

4. Line 326: What is Table ?? meaning for?.

Response: This has been resolved. The table has been correctly referenced.

5. The authors can add some discussion on the potential application prospect on environmental management and monitoring.

Response: Some discussions have been added on the application of aerosol scattering on environmental management and monitoring in the introductory section and highlighted in blue.

Thank you once more for your significant contribution to improving the quality of our paper.

Reviewer 3 Report

1. The comparison and validation of the GRAP profile have been improved.

2.Figure 2: Please give numbers or labels to each sub-figure, like (a)(b).

No further comment.

Author Response

Thank you very much for taking the time to review our paper. The issues you raised have been addressed as follows;

2. Figure 2: Please give numbers or labels to each sub-figure, like (a)(b).

In Figure 2, like in Figure 6, I have used the titles of the sub-figures as references instead of the dummy (a), (b), etc.

Thank you once more for your significant contribution to improving our paper's quality.